# Analysis of Cellular Stress Assay Parameters and Intracellular ATP in Platelets: Comparison of Platelet Preparation Methods

**DOI:** 10.3390/ijms25094885

**Published:** 2024-04-30

**Authors:** Belay Tessema, Janine Haag, Ulrich Sack, Brigitte König

**Affiliations:** 1Institute of Clinical Immunology, Faculty of Medicine, University of Leipzig, 04103 Leipzig, Germany; ulrich.sack@medizin.uni-leipzig.de; 2Magdeburg Molecular Detections GmbH & Co. KG, 39104 Magdeburg, Germany; mmd@mmd-web.de (J.H.); brigitte.koenig@medizin.uni-leipzig.de (B.K.); 3Department of Medical Microbiology, College of Medicine and Health Sciences, University of Gondar, Gondar P.O. Box 196, Ethiopia; 4Institute of Medical Microbiology and Virology, Faculty of Medicine, University of Leipzig, 04103 Leipzig, Germany

**Keywords:** platelets, cellular stress assay, ATP, platelet isolation

## Abstract

Platelets are metabolically active, anucleated and small circulating cells mainly responsible for the prevention of bleeding and maintenance of hemostasis. Previous studies showed that platelets mitochondrial content, function, and energy supply change during several diseases such as HIV/AIDS, COVID-19, pulmonary arterial hypertension, and in preeclampsia during pregnancy. These changes in platelets contributed to the severity of diseases and mortality. In our previous studies, we have shown that the seahorse-based cellular stress assay (CSA) parameters are crucial to the understanding of the mitochondrial performance in peripheral blood mononuclear cells (PBMCS). Moreover, the results of CSA parameters were significantly influenced by the PBMC preparation methods. In this study, we assessed the correlation of CSA parameters and intracellular ATP content in platelets and evaluated the effects of platelet preparation methods on the results of CSA parameters and intracellular ATP content. We compared the results of CSA parameters and intracellular ATP content in platelets isolated by density centrifugation with Optiprep and simple centrifugation of blood samples without Optiprep. Platelets isolated by centrifugation with Optiprep showed a higher spare capacity, basal respiration, and maximal respiration than those isolated without Optiprep. There was a clear correlation between basal respiration and maximal respiration, and the whole-ATP content in both isolation methods. Moreover, a positive correlation was observed between the relative spare capacity and whole-cell ATP content. In conclusion, the results of seahorse-based CSA parameters and intracellular ATP content in platelets are markedly influenced by the platelet isolation methods employed. The results of basal respiration and maximal respiration are hallmarks of cellular activity in platelets, and whole-cell ATP content is a potential hint for basic platelet viability. We recommend further studies to evaluate the role of CSA parameters and intracellular ATP content in platelets as biomarkers for the diagnosis and prediction of disease states.

## 1. Introduction

Platelets are small, anucleated, and short-lived (7–10 days) circulating cells mainly responsible for the prevention of bleeding and maintenance of hemostasis [1,2]. Generally, platelets contain only 5–8 mitochondria per cell, while most human nucleated cells contain hundreds of mitochondria [3]. However, platelets are metabolically active and have much higher levels of ATP turnover compared to resting mammalian muscle cells [4]. In platelets, this energy demand is met using a metabolic system that combines the efforts of glycolysis and mitochondrial oxidative phosphorylation (OXPHOS). OXPHOS provides 30–40% of cellular ATP, while glycolysis provides the remaining 60% of cellular ATP in platelets at rest [5]. Interestingly, to meet this energy demand with so few mitochondria, platelets have shown metabolic flexibility; activated platelets show a glycolytic phenotype even as they preserve mitochondria function [6] The resting platelets actively generate and utilize glycogen under normoglycemic conditions. In resting platelets, there is a steady-state turnover of glycogen that is shifted to net degradation/consumption upon activation [7]. This metabolic flexibility helps platelets to adapt to different conditions, such as hypoxia or the presence of agents that inhibit mitochondrial functions [8]. Platelets may provide an easy-to-harvest, real-time window into the metabolic shift occurring in the body and represent a useful surrogate for interrogating the glycolytic shift central to a disease condition [9]. Therefore, platelets could be considered as an option in mitochondria fitness studies other than peripheral blood mononuclear cells (PBMCs).

In contrast to other immune cells, platelets do not have a nucleus [10]. Therefore, platelets are not able to genetically regulate situations such as stress or altered fuel supply. Stress resistance and adjustment to available fuels are primarily regulated by enzyme activity. This implies damage in platelets or malfunctions has a higher chance of being conserved in the cells and therefore in the metabolic performance [11,12]. This fact is the basis for using platelets as alternative cell types in this study for the analysis of cellular stress and Intracellular ATP content considering them as a mirror of the situation within the body. 

Many diseases, such as cancer, fatigue, and neurodegenerative disorders, as well as situations such as poor performance in sports or aging, are connected to weak energy supply [13,14,15,16]. The diagnosis of the potential involvement of mitochondria in such conditions has become easier with the invention of the Seahorse Analyzer [11,16]. Our previous studies and a study by Hill et al. have shown that the CSA parameters and mitochondrial mass are key to the complete understanding of the mitochondrial performance in PBMCs [17,18,19].

It became clear that the mitochondrial spare capacity defined as the difference between basal and maximal respiration is one of the most important parameters of the Seahorse-based CSA. Other parameters, such as proton leak, non-mitochondrial respiration, or coupling efficiency are likely not that important for the assessment of the bioenergetic performance of PBMCs. Nevertheless, these parameters might be of importance in other cell types such as in platelets or certain diseases. A previous study showed that long-term treated *Human immunodeficiency virus*/acquired immunodeficiency syndrome (HIV/AIDS) is associated with reduced platelet mitochondrial content, platelet mitochondrial dysfunction, and reduced energy supply [20]. A study by Yasseen et al. also suggests that hyperactive platelets with impaired exocytosis may be integral parts of the pathophysiology dictating severity and mortality in COVID-19 patients [21]. Similarly, in preeclampsia, there exists either a loss or early reversal of a normal biologic mechanism of platelet mitochondrial respiratory reduction associated with normal pregnancy [22]. 

In our previous studies, we found that extracellular acidification is a potential marker for age-dependent alterations in cellular metabolism, and hence it is an important parameter that should be considered when diagnosing the bioenergetic performance of PBMCs [18,19]. In our previous studies, we also demonstrated that the results of the seahorse-based CSA parameters in PBMCs are significantly influenced by the isolation methods employed [18,19]. Consequently, we hypothesized that the platelet isolation methods could also influence the results of seahorse-based CSA parameters and intracellular ATP content in platelets. Therefore, the main objectives of this study were to assess the correlation of seahorse-based CSA parameters and ATP content in platelets and to evaluate the effects of the two commonly used platelet preparation methods by centrifugation with Optiprep and without Optiprep on the results of CSA parameters and intracellular ATP content in platelets. 

## 2. Results

### 2.1. Comparison of CSA Parameters in Platelets Isolated with and without Optiprep

Platelets isolated with Optiprep displayed a much higher basal and maximal respiration. Furthermore, after inhibition of the mitochondria with Oligomycin, the extracellular acidification (ECAR) is increased in platelets isolated with Optiprep. Most importantly, the maximal respiration in platelets isolated without Optiprep is comparable to the basal respiration. This finding implies that platelets isolated with Optiprep display a much higher spare capacity than those isolated without Optiprep (Figure 1).

The amount of whole-cell ATP did not show marked differences in platelets isolated with and without Optiprep methods (with optiprep ATP = 140.21 ± 55.28 amol/cell; without Optiprep ATP = 136.28 ± 60.35 amol/cell). The results of basal respiration, basal OCR, non-mitochondrial respiration (pmol/min), maximal respiration, and extracellular acidification are in both cases approximately normally distributed (Shapiro-Wilk (W) ≈ 1). Interestingly, the W values for spare capacity data showed marked differences in the two platelet isolation methods. The W value for spare capacity is close to one in platelets isolated with Optiprep indicating normal distribution and higher than the W value in platelets isolated without Optiprep (Table 1). 

The effect size of the two platelet isolation methods on the results of CSA parameters was determined using Cohens d test and the d values were interpreted as d > 0.8 = strong effect, d > 0.5 = medium effect, and d > 0.2 = small effect. As expected, the strongest differences between platelets isolated with and without optiprep were observed in basal respiration (d = 0.81) and maximal respiration (d = 1.43). In accordance with the differences in maximal respiration, the spare capacity (pmol/min: d = 1.64; %: d = 1.73) and BHI (d = 1.41) showed significant differences. These observations revealed that platelets isolated in the presence of optiprep were activated. Moreover, the isolation methods have medium effect on the results of Basal OCR (d = 0.80), non-mitochondrial respiration [%] (d = 0.53), and maximal ECAR (FCCP) [mpH/min] (d = 0.70). However, the two isolation methods do not affect the results of proton leak [pmol/min] and basal ECAR [mpH/min] (Table 2).

### 2.2. Correlation of CSA and ATP Content in Platelets Isolated with and without Optiprep Methods

As shown in Figure 2, there are correlations between OCR-related parameters, ECAR-related parameters, and ATP-related parameters. There is also a correlation between ECAR- and OCR-related parameters and even between OCR- or ECAR-related parameters and ATP-related parameters. These correlations are probably because cells with higher oxygen consumption also tend to have higher acidification or ATP content. This study demonstrated that spare capacity, whole-cell ATP and non-mitochondrial respiration are potential hallmarks of bioenergetic fitness in platelets. The correlations of CSA parameters and ATP measurements in platelets isolated with and without optiprep methods are summarized using a heatmap correlation matrix in Figure 2.

It is of interest whether parameters in one group (ECAR, OCR, or ATP) also correlate with parameters of other groups in platelets isolated with and without Optiprep methods. As shown in Figure 3, there is a positive correlation between whole-cell ATP (amol/cell) and spare capacity (%) in both platelets isolated with and without Optiprep. The correlation coefficient of spare capacity and whole-cell ATP in platelets isolated with Optiprep was 0.593 while for platelets isolated without Optiprep was 0.444. These findings are under the assumption that a higher spare capacity might lead to higher cellular ATP content and therefore to better cellular fitness. Nevertheless, some platelets isolated without Optiprep show a spare capacity of zero % (Figure 3B). These could have limited the correlation between whole-cell ATP and spare capacity in platelets isolated without Optiprep below 0.5. In this context, it should be mentioned that the correlation coefficient of basal respiration and whole-cell ATP in platelets isolated with Optiprep was 0.649 while for platelets isolated without Optiprep was 0.616 (Figure 3). 

This study also demonstrated negative correlations between non-mitochondrial respiration and basal respiration, maximal respiration, and spare capacity in platelets isolated by centrifugation with Optiprep as well as without Optiprep methods. The correlation coefficient between non-mitochondrial respiration and basal respiration in platelets isolated with Optiprep was −0.418 while without Optiprep was −0.719. Likewise, the correlation coefficient between non-mitochondrial respiration and maximal respiration was −0.458 in platelets isolated with Optiprep and −0.637 in platelets isolated without Optiprep. Between non-mitochondrial respiration and basal respiration, the correlation coefficient in platelets isolated with Optiprep was −0.446 and −0.35 in platelets isolated without Optiprep (Figure 4). 

In nucleated immune cells, non-mitochondrial respiration is mainly the result of surface respiration or the activity of chaperones within the endoplasmic reticulum. These mechanisms, however, are probably poorly developed within platelets and even the endoplasmic reticulum is only residually present. Therefore, the observed non-mitochondrial respiration is potentially related to the reactive oxygen species (ROS) generating processes located within the mitochondria. Therefore, an increase in non-mitochondrial respiration is a potential indicator of a decline in mitochondrial fitness.

## 3. Discussion

In this study, we carried out a comprehensive analysis of seahorse-based cellular stress assay parameters and intracellular ATP measurements in platelets isolated with and without Optiprep. This study showed that platelets isolated using Optiprep have a higher spare capacity than those isolated solely by centrifugation in the absence of Optiprep. Moreover, the spare capacity in platelets isolated without Optiprep was only slightly higher than the actual basal respiration. Similarly, Cohen’s d-test results showed that the strongest differences between platelets isolated with and without Optiprep were observed in basal respiration and maximal respiration. In accordance with the differences in maximal respiration, the spare capacity and BHI showed significant differences. We, therefore, conclude that Optiprep activates platelets when platelets are isolated using centrifugation with Optiprep. Activated platelets are usually considered in contrast to resting platelets. Similarly, a previous study showed that the activation of platelets leads to an increase in spare capacity [12]. 

This study also demonstrated that platelets isolated with Optiprep displayed higher basal respiration and maximal respiration. The amount of whole-cell ATP did not show marked differences in platelets isolated with and without Optiprep methods. Nevertheless, there was a clear correlation between basal respiration and maximal respiration, and the whole-ATP content in both isolation methods. These findings imply that basal respiration and maximal respiration are indeed hallmarks of cellular activity, and whole-cell ATP is a potential hint for basic platelet viability. This finding was also supported by a positive correlation between the relative spare capacity and whole-cell ATP observed in this study. 

Interestingly, non-mitochondrial respiration seems to be a potential reason for decreased mitochondrial activity in platelets. The non-mitochondrial activity showed a negative correlation with basal and maximal respiration, as well as spare capacity. A significant impact of the non-mitochondrial respiration on the cellular ATP content, however, was not observed. This finding challenges the previous observations on the role of the proton leak at least tested in this manner on mitochondrial health. In our previous studies, we observed that the proton leak had none or only little effect on the PBMCs viability [18,19]. The data of proton leak can be considered as a non-normal distribution except in pmol/min in Optiprep-isolated platelets. This shows the proton leak is a highly controversial parameter that should be interpreted with caution. The non-mitochondrial respiration, in contrast, is statistically much more reliable and might, therefore, serve as a trustworthy parameter for the assessment of oxidative stress, at least in platelets. 

In this study, the results of seahorse-based CSA and ATP content in platelets were not significantly influenced by potential factors namely the age and gender of the study participants as well as the seasons of blood collection. However, the previous studies demonstrated significant effects of age, gender, and season on the results of CSA parameters in PBMCs. The effects of these factors on the results of CSA in PBMCs were explained as possible seasonal changes in the well-being of the individuals such as vitamin D deficiency during winter and age-related decline in physiological and biochemical fitness [18,19,23,24]. In platelets, the absence or only little effects of these factors on the results of CSA parameters could be a low platelet variability during these conditions. 

The presence of a nucleus in PBMCs helps them to use a broad range of regulatory mechanisms during oxidative stress. On the contrary, the absence of nucleus in platelets probably leads to cell death during weak cellular integrity caused by oxidative stress [10]. Thus, in comparison to other sources of mitochondria, platelets are more regenerative. In this regard, platelets are considered as alternative cells that reflect the bioenergetic function of muscles, brains, and other organs [11] and can be used as a biomarker for systemic mitochondrial dysfunction in the body [25]. 

Limitations: In this study, we did not analyze the molecular mechanisms of changes in cellular stress assay parameters and intracellular ATP in platelets. Moreover, the comparisons of CSA results from platelets isolated by the two platelet isolation methods, centrifugation with and without Optiprep were made without matching the clinical and socio-demographic characteristics of study participants such as gender, age, and season of blood sample collection. These unmatched characteristics could have affected the observed results of CSA between the two platelet isolation methods. However, the findings of this study are strengthened by using well-standardized protocols for seahorse-based CSA analysis and intracellular ATP measurements in platelets. 

## 4. Materials and Methods

### 4.1. Study Design, Period and Settings

A comparative retrospective cross-sectional study was carried out at the Magdeburg Molecular Detections (MMD) laboratory between October 2017 and February 2020. Healthy individuals and patients who gave blood for CSA and intracellular ATP measurements during the study period were enrolled in this study. The socio-demographic characteristics data of study participants such as gender, age and season of blood sample collection, platelet isolation methods used, and CSA parameters and intracellular ATP measurement results were collected from the MMD laboratory records. 

### 4.2. Study Participants

A total of 125 study participants, 65 males and 60 females were investigated in this study. The age range of study participants was 10 to 90 years old. The platelets of 75 participants were isolated via centrifugation in the absence of Optiprep (without Optiprep) and the platelets of 50 patients were isolated by density centrifugation using Optiprep (with Optiprep). The CSA of 100 participants was carried out in a Seahorse XFp Analyzer (Agilent^TM^, Santa Clara, CA, USA) and the remaining 25 persons were analyzed using a Seahorse XF96 Analyzer (Agilent^TM^, Santa Clara, CA, USA). 

### 4.3. Blood Sample Collection and Platelet Isolation

From each participant, up to 16 mL venous blood was collected aseptically in a vacutainer tube with Citrate-phosphate-dextrose solution with adenine (CPDA) anti-coagulant. The collected blood samples were delivered to MMD laboratory immediately at room temperature and platelets were isolated for CSA and Intracellular ATP measurement within 24 h of blood collection. For validation issues we assessed the CSA changes during storage of the identical sample of whole blood for 0, 24, 48 and 72 h after collection. Up to 48 h after blood draw no significant changes in CSA parameters were observed. 

Platelet isolation without Optiprep was conducted as follows: first, 8 mL of whole blood was centrifuged at 200× *g* for 10 min (full break), the platelet-rich supernatant was added to a fresh 15 mL tube and filled up to 10 mL with RPMI-1640 full medium (RPMI-1640 medium by Sigma Aldrich supplemented with 10 mM glucose and 2 mM pyruvate, both purchased from CarlRoth^®^, Karlsruhe, Germany). After 10 min of incubation, the platelets were centrifuged at 2200× *g* for 16 min with a full brake (washing step 1). The washing step was repeated (washing step 2) [26]. Subsequently, the supernatant was again discarded, the pellet was diluted and filled up to 10 mL with RPMI-1640 medium and PGI2 at a final concentration of 0.5 µM was added. 

The platelet isolation via density centrifugation with optiprep was done using Optiprep (StemcellTM technologies, Vancouver, Canada) from 8–16 mL of blood according to the supplier’s recommendation and following previously published protocols [27,28]. After the centrifugation step with Optiprep (1.063 g/cm^3^), the whole soluble fractions were collected, and the cell pellet was discarded. Subsequently, the soluble fraction was filled up to 10 mL with RPMI-1640 full medium. After 10 min of incubation, the platelets were centrifuged at 2200× *g* for 16 min with a full brake (washing step 1). The washing step was repeated (washing step 2). Subsequently, the supernatant was again discarded, the pellet was diluted and filled up to 10 mL with RPMI-1640 medium and PGI2 at a final concentration of 0.5 µM was added. 

After completing all washing steps of both platelet isolation methods, the cell number was determined either microscopically using trypan blue-stained cells in a C-Chip hemocytometer (Neubauer improved by NanoEnTek) under an AE31 trinocular (Motic, Wetzlar, Germany) or photometrically using transparent 96-well plates (Greiner bio-one, Monroe, NC, USA) in a Mithras LB 940 (Berthold Technologies, Bad Wildbad, Germany).

### 4.4. Cellular Stress Assay (CSA)

The CSA was conducted using the Seahorse XFp Analyzer (Agilent^TM^, Santa Clara, CA, USA) or the Seahorse XF96 Analyzer (Agilent^TM^, Santa Clara, CA, USA) according to the manufacturer’s protocol in triplicates as described previously with some modifications [17]. The following modifications were made in this study, 3 × 10^6^ platelets per well (100 µL) and 1 µM FCCP were used in the protocol. In brief, the Seahorse-based CSA parameters analysis is done with sequential stepwise injection of various mitochondrial inhibitors as described in our previous studies [18,19]. Mitochondrial inhibitors used in this study were carbonyl cyanide 4-(trifluoromethoxy) phenylhydrazone (FCCP), Oligomycin, and Rotenone and Antimycin A. This sequential procedure enables the measurement of CSA parameters such as basal respiration, proton leak, spare capacity, maximal respiration, non-mitochondrial respiration, and extracellular acidification rate.

### 4.5. Measurement of Intracellular ATP

To measure the amount of intracellular ATP, isolated platelets were seeded into white plates µCLEAR, White Cellstar^®^ (Greiner bio-one, Frickenhausen, Germany) with 37,500 platelets (50 µL) per well. Subsequently, inhibitors were added in the same concentrations as used in Seahorse measurements and the wells were filled up to a final volume of 100 µL. After 18 min of incubation at 37 °C (without CO_2_), 100 µL of CellTiter-Glo^®^ (Promega, Walldorf, Germany) was added for another 10 min at room temperature without light exposure. Eventually, the luminescence was measured as a singlet using the Mithras LB 940 (Berthold Technologies, Bad Wildbad, Germany) according to the manufacturer’s protocol.

### 4.6. Statistical Analyses

The Seahorse CSA parameters data were analyzed with Wave 2.6.0 software (Agilent, Santa Clara, CA, USA). The analyzed CSA data were transferred into Microsoft Excel for data cleaning to avoid errors in data entry. Then, the cleaned data were transferred into R program and the statistical analyses were performed with R program (version 3.6.2) and RStudio (version 1.2.5033). For statistical analysis, we used the following packages: tidyverse, ggpubr, table1, formattable, summarytools, caret, psych, MASS, lsr, gmodels, and broom. The data distribution was assessed by Shapiro-Wilk test. The correlation of CSA parameters and intracellular ATP content is presented using a heatmap correlation matrix figure. To assess the possible influence of sample sizes on *p*-values, we used the Cohens d test to determine the effect size of our observations. 

## 5. Conclusions

In this study, platelets isolated using centrifugation with Optiprep showed a higher spare capacity, basal respiration, and maximal respiration than those isolated without Optiprep. There was also a clear correlation between basal respiration and maximal respiration, and the whole-ATP content in both isolation methods. Moreover, a positive correlation was observed between the relative spare capacity and whole-cell ATP content. The non-mitochondrial activity showed negative correlations with basal and maximal respiration and spare capacity. In platelets, non-mitochondrial respiration seems to be a potential marker for decreased mitochondrial activity. We recommend further studies to elucidate the role of CSA parameters and intracellular ATP content in platelets as biomarkers for the diagnostic and therapeutic evaluation of different diseases.

## Figures and Tables

**Figure 1 ijms-25-04885-f001:**
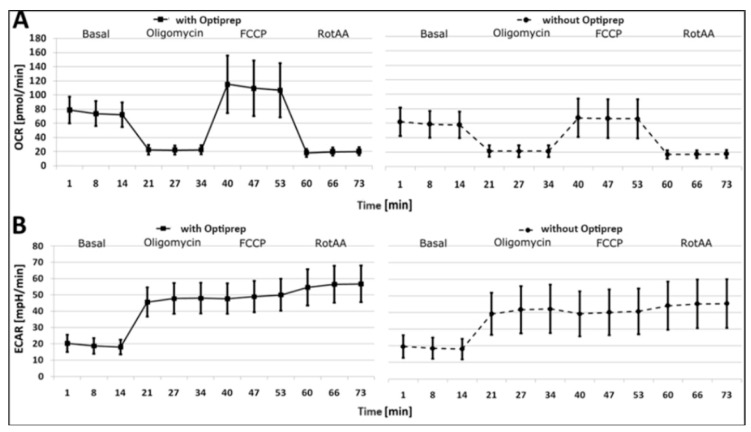
Cellular stress assays (CSA) parameters in platelets isolated with Optiprep (n = 50) or without Optiprep (n = 75). (**A**) Oxygen consumption rate (OCR) and (**B**) extracellular acidification rate (ECAR). The results show the mean ± standard deviation of OCR and ECAR results.

**Figure 2 ijms-25-04885-f002:**
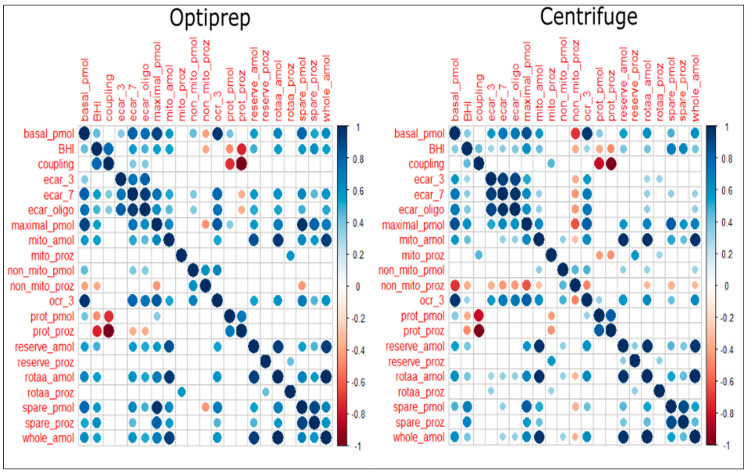
Heatmap correlation matrix of CSA parameters and ATP measurements in platelets isolated either with (n = 50) or without Optiprep (n = 75). Basal_pmol = basal respiration (pmol/min), BHI = bioenergetic health index, coupling = coupling efficiency (%), ecar_3 = basal ECAR (mpH/min), ecar_7 Correlation matrix of CSA parameters and ATP measurements in platelets isolated either with (n = 50) or without Optiprep (n = 75). Basal_pmol = basal respiration (pmol/min), BHI = bioenergetic health index, coupling = coupling efficiency (%), ecar_3 = basal ECAR (mpH/min), ecar_7 = ECAR after FCCP (mpH/min), ecar_oligo = ECAR after Oligomycin (mpH/min), maximal_pmol = maximal respiration (pmol/min), mito_amol = intracellular ATP after 2-DG (amol/cell), mito_proz = intracellular ATP after 2-DG (%), non_mito_pmol = non-mitochondrial respiration (pmol/min), non_mito_proz = non-mitochondrial respiration (%), ocr_3 = basal OCR (pmol/min), prot_pmol = proton leak (pmol/min), prot_proz = proton leak (%), reserve_amol = intracellular ATP after 2-DG and RotAA (amol/cell), reserve_proz = intracellular ATP after 2-DG and RotAA (%), rotaa_amol = intracellular ATP after RotAA (amol/cell), reserve_proz = intracellular ATP after RotAA (%), spare_pmol = spare capacity (pmol/min), spare_proz = spare capacity (%), whole_amol = whole-cell ATP (amol/cell).

**Figure 3 ijms-25-04885-f003:**
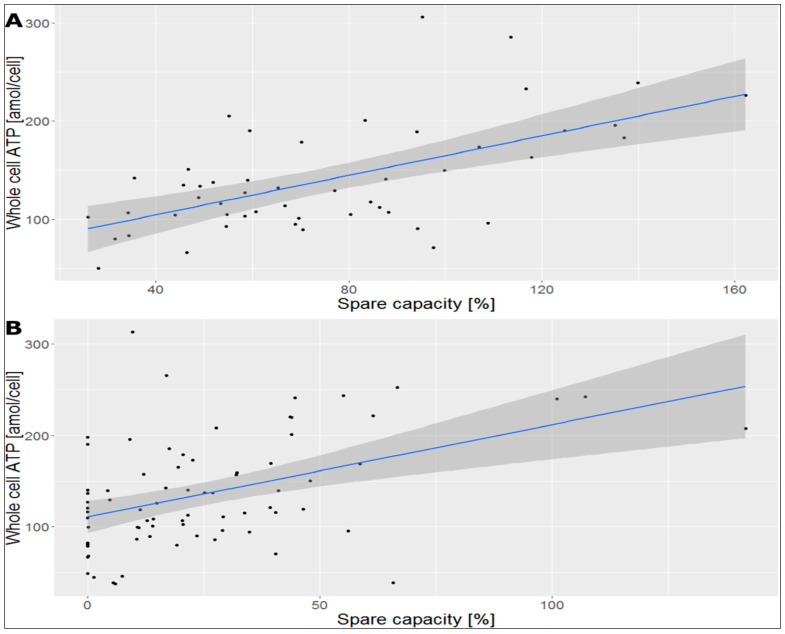
Correlation diagrams of whole-cell ATP (amol/cell) and spare capacity (%) in platelets isolated with Optiprep ((**A**); n = 50) and platelets isolated without Optiprep ((**B**); n = 75). The correlation coefficient of spare capacity and whole-cell ATP in platelets isolated with Optiprep (**A**) was 0.593 while for platelets isolated without Optiprep (**B**) was 0.444.

**Figure 4 ijms-25-04885-f004:**
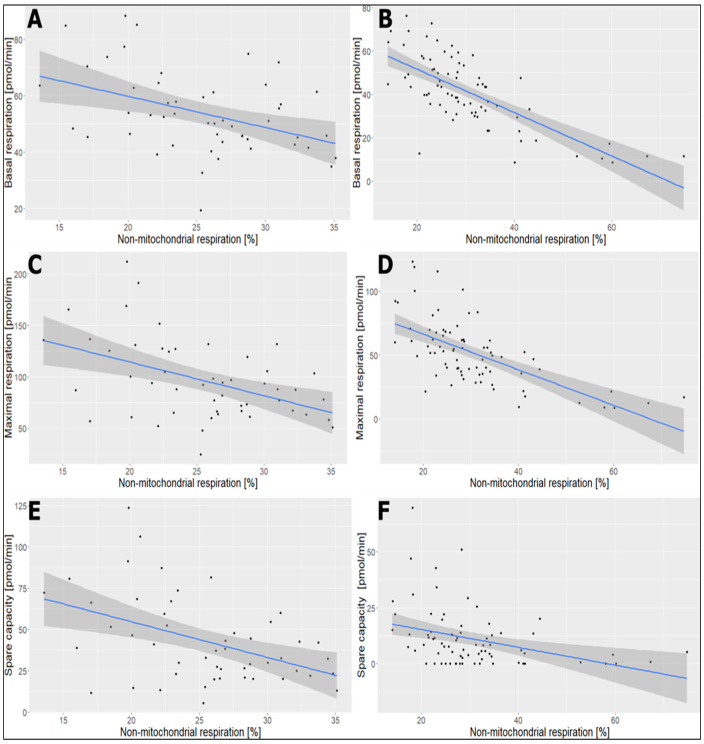
Correlation diagrams of (%) and basal respiration in platelets isolated with Optiprep (**A**) and without Optiprep (**B**), non-mitochondrial respiration and maximal respiration in platelets with Optiprip (**C**) and without Optiprip (**D**), non-mitochondrial respiration and spare capacity in platelets isolated with Optiprep (**E**) and without Optiprep (**F**). Platelets isolated with Optiprep (n = 50) and platelets isolated without Optiprep (n = 75). Correlation coefficient: (**A**) = −0.418; (**B**) = −0.719; (**C**) = −0.458; (**D**) = −0.637; (**E**) = −0.446; (**F**) = −0.35.

**Table 1 ijms-25-04885-t001:** The results of cellular stress assays parameters obtained from platelets isolated with Optiprep (n = 50) and without Optiprep (n = 75).

Parameter	With Optiprep			Without Optiprep		
	Mean (SD)	95% CI	Shapiro-Wilk (W)	Mean (SD)	95% CI	Shapiro-Wilk (W)
Basal respiration (pmol/min)	51.8 (15.6)	47.5–56.2	0.968	41.1 (16.4)	37.4–44.8	0.982
basal OCR (pmol/min)	72.0 (17.5)	67.1–76.8	0.971	57.6 (18.3)	53.5–61.8	0.984
non-mitochondrial respiration (pmol/min)	18.1 (5.5)	16.6–19.7	0.970	16.5 (5.6)	15.3–17.8	0.956
non-mitochondrial respiration (%)	25.4 (5.4)	23.9–26.9	0.982	30.5 (11.8)	27.9–33.2	0.859
coupling efficiency (%)	93.0 (6.0)	91.3–94.7	0.740	89.6 (13.2)	86.6–92.6	0.894
proton leak (pmol/min)	3.7 (2.3)	3.1–4.3	0.954	4.5 (5.4)	3.3–5.7	0.793
proton leak (%)	7.2 (5.8)	5.6–8.8	0.699	10.9 (12.4)	8.1–13.7	0.817
maximal respiration (pmol/min)	97.0 (38.7)	86.2–108.0	0.943	51.6 (25.8)	45.8–57.5	0.962
spare capacity (pmol/min)	43.1 (26.3)	35.8–50.4	0.919	11.1 (13.2)	8.2–14.1	0.777
spare capacity (%)	75.6 (32.5)	66.5–84.6	0.955	25.3 (26.5)	19.3–31.3	0.823
bioenergetic health index (BHI)	1.5 (0.4)	1.4–1.6	0.955	0.7 (0.6)	0.6–0.9	0.898
basal ECAR (mpH/min)	18.0 (4.5)	16.7–19.2	0.980	18.0 (6.2)	16.6–19.3	0.972
ECAR Oligomycin (mpH/min)	48.0 (9.4)	45.4–50.6	0.982	42.2 (14.6)	38.9–45.5	0.956
maximal ECAR (FCCP) (mpH/min)	47.7 (9.3)	45.1–50.3	0.993	39.2 (13.6)	36.2–42.3	0.964

OCR = Oxygen consumption rate; ECAR = Extracellular acidification rate; FCCP = carbonyl cyanide 4-(trifluoromethoxy) phenylhydrazone; SD = standard division; CI = confidence interval; W interpreted as follows: if the W values are close to 1, the data is probably normally distributed. If the W values are far from 1, the assumption of normal distribution of the data is probably not met.

**Table 2 ijms-25-04885-t002:** The results of the Cohens d based on cellular stress assay measurements in platelets isolated with or without Optiprep.

Parameter	Cohens d *	Interpretation
basal respiration [pmol/min]	0.81	strong
basal OCR [pmol/min]	0.80	medium
non-mitochondrial respiration [pmol/min]	0.29	small
non-mitochondrial respiration [%]	0.53	medium
coupling efficiency [%]	0.31	small
proton leak [pmol/min]	0.18	no
proton leak [%]	0.37	small
maximal respiration [pmol/min]	1.43	strong
spare capacity [pmol/min]	1.64	strong
spare capacity [%]	1.73	strong
bioenergetic health index (BHI)	1.41	strong
basal ECAR [mpH/min]	0.00	no
ECAR Oligomycin [mpH/min]	0.45	small
maximal ECAR (FCCP) [mpH/min]	0.70	medium

Data were collected from 50 platelet preparations with Optiprep and 75 platelet preparations without Optiprep. OCR = Oxygen consumption rate; ECAR = Extracelluar acidification rate; FCCP = carbonyl cyanide 4-(trifluoromethoxy) phenylhydrazone; SD = standard division; CI = confidence interval. * = Cohens d result shows the effect size of the two platelet isolation methods and interpreted as d > 0.2 = small effect, d > 0.5 = medium effect, and d > 0.8 = strong effect.

## Data Availability

All relevant data are included in the paper. The data are available on request from the corresponding author.

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
