# Peer review of "Analysis of Cellular Stress Assay Parameters and Intracellular ATP in Platelets: Comparison of Platelet Preparation Methods"

_ijms, 2024, doi:10.3390/ijms25094885_

Round 1
Reviewer 1 Report
Comments and Suggestions for Authors
The manuscript entitled " Analysis of cellular stress assay parameters and intracellular
ATP in platelets: comparison of platelet preparation methods” by Belay Tess et al. presents the results of a specialized study demonstrating that using optiprep, rather than solely centrifugation, is a more effective method for preserving intracellular ATP during platelet isolation. While the paper is interesting, it seems somewhat speculative, given that centrifugation is known to potentially activate platelets and induce granule secretion.
Major concerns:
- Have the authors considered measuring secreted ATP from platelets to validate their results? Additionally, they should explain the meaning of "spare capacity."
- The paragraph stating, "We, therefore, conclude that optiprep activates platelets when platelets are isolated using centrifugation with optiprep. Similarly, a previous study showed that the activation of platelets leads to an increase in spare capacity," is contradictory. Please review the wording.
- The authors should explain the results of the Cohen's d statistic to help readers understand Table 2, particularly those who are not experts in the field.
- The correlation diagrams Figure 3 and 4, should be better explained to visualize the results clearly. Additionally, the descriptions are not clear and need to be improved for better understanding.
Author Response
Reviewer 1 Report 1
Review Report Form
Open Review
(x) I would not like to sign my review report
( ) I would like to sign my review report
Quality of English Language
( ) I am not qualified to assess the quality of English in this paper
( ) English very difficult to understand/incomprehensible
( ) Extensive editing of English language required
( ) Moderate editing of English language required
( ) Minor editing of English language required
(x) English language fine. No issues detected
|
Yes |
Can be improved |
Must be improved |
Not applicable |
|
|
Does the introduction provide sufficient background and include all relevant references? |
(x) |
( ) |
( ) |
( ) |
|
Are all the cited references relevant to the research? |
(x) |
( ) |
( ) |
( ) |
|
Is the research design appropriate? |
(x) |
( ) |
( ) |
( ) |
|
Are the methods adequately described? |
( ) |
(x) |
( ) |
( ) |
|
Are the results clearly presented? |
( ) |
( ) |
(x) |
( ) |
|
Are the conclusions supported by the results? |
(x) |
( ) |
( ) |
( ) |
Comments and Suggestions for Authors
General Comment: The manuscript entitled " Analysis of cellular stress assay parameters and intracellular ATP in platelets: comparison of platelet preparation methods” by Belay Tess et al. presents the results of a specialized study demonstrating that using optiprep, rather than solely centrifugation, is a more effective method for preserving intracellular ATP during platelet isolation. While the paper is interesting, it seems somewhat speculative, given that centrifugation is known to potentially activate platelets and induce granule secretion.
Response: Dear reviewer, we very much appreciate your review of our work and valuable comments and suggestions for improvement of our manuscript. We have revised our manuscript considering your comments and suggestions to improve the quality of our work.
We agree that centrifugation is known to potentially activate platelets. However, one of the objectives of this study was to compare the effect of the two commonly used platelet preparation methods namely centrifugation with Optiprep and centrifugation without Optiprep on the results of CSA parameters and intracellular ATP content in platelets. This study clearly showed that platelets isolated using centrifugation with Optiprep have higher basal respiration, maximal respiration, and spare capacity than those isolated solely by centrifugation without Optiprep. It implies that Optiprep activates platelets when platelets are isolated using centrifugation with Optiprep. The intracellular ATP content did not show marked differences in platelets isolated with and without Optiprep methods. Nevertheless, there was a clear correlation between basal respiration and maximal respiration, and the intracellular ATP content in both platelet isolation methods.
Major concerns:
Comment 1: Have the authors considered measuring secreted ATP from platelets to validate their results? Additionally, they should explain the meaning of "spare capacity."
Response: We measured the intracellular ATP content in platelets isolated with and without Optiprep methods at basal state (without the addition of FCCP). There was no significant difference. “Activation” of platelets is not the same as increased ATP levels or ATP release. Nonetheless, we analyzed for the presence of ATP in the medium – no difference in extracellular ATP was detected between both isolation methods. Activation of platelets leads to a higher “spare capacity” which is defined as the difference between basal and maximal possible respiration (see page 2, lines 80-81).
Comment 2: The paragraph stating, "We, therefore, conclude that optiprep activates platelets when platelets are isolated using centrifugation with optiprep. Similarly, a previous study showed that the activation of platelets leads to an increase in spare capacity," is contradictory. Please review the wording.
Response: In this paragraph, our conclusion Optiprep activates platelets when platelets are isolated using centrifugation with Optiprep is based on an increase in spare capacity, similarly, the previous study showed that the activation of platelets leads to an increase in spare capacity. So we believe that it is a similar finding not contradictory.
Comment 3: The authors should explain the results of the Cohen's d statistic to help readers understand Table 2, particularly those who are not experts in the field.
Response: The results of the Cohen´s d statistic are now explained in detail. Pages 4 and 5, lines 140-150.
Comment 4: The correlation diagrams Figure 3 and 4, should be better explained to visualize the results clearly. Additionally, the descriptions are not clear and need to be improved for better understanding.
Response: The correlation diagrams in Figures 3 and 4 are explained in detail and the descriptions of the figures are improved. Pages 6 - 8, lines 207 - 265.
Reviewer 2 Report
Comments and Suggestions for Authors
Author Response
Reviewer 2 Report 1
Review Report Form
Open Review
(x) I would not like to sign my review report
( ) I would like to sign my review report
Quality of English Language
( ) I am not qualified to assess the quality of English in this paper
( ) English very difficult to understand/incomprehensible
( ) Extensive editing of English language required
( ) Moderate editing of English language required
( ) Minor editing of English language required
(x) English language fine. No issues detected
|
Yes |
Can be improved |
Must be improved |
Not applicable |
|
|
Does the introduction provide sufficient background and include all relevant references? |
( ) |
( ) |
(x) |
( ) |
|
Are all the cited references relevant to the research? |
(x) |
( ) |
( ) |
( ) |
|
Is the research design appropriate? |
( ) |
(x) |
( ) |
( ) |
|
Are the methods adequately described? |
( ) |
( ) |
(x) |
( ) |
|
Are the results clearly presented? |
( ) |
(x) |
( ) |
( ) |
|
Are the conclusions supported by the results? |
( ) |
( ) |
(x) |
( ) |
Comments and Suggestions for Authors
General Comment: Tessema et al. investigated blood samples collected from 125 subjects within about 2.5 years until February 2020 to analyze the platelet preparations by the seahorse-based cellular stress assay (CSA) and to quantify the intracellular ATP content by chemiluminescence. They compared two platelet preparation methods, i. e. conventionally washed platelets and the density centrifugation with Optiprep. They observed both a significant effect of the preparation on their analytical parameters and significant correlations between the CSA parameters and the whole cell ATP-content. The CSA approach to assess the bioenergetics of blood cells is intriguing and may indeed deserve wider attention.
Response: Dear reviewer, we are very much thankful for your meticulous review of our work and for providing us with very useful comments and suggestions. The manuscript has been revised according to your comments and suggestions to improve its quality.
Comment 1: In their conclusion, they “recommend further studies to elucidate the role of CSA parameters and intracellular ATP content in platelets as rapid and precise biomarkers for the diagnostic and therapeutic evaluation of different disease”. Corresponding data on the minimal time required from sampling to the final CSA results and on the reproducibility of the method should be presented to support this important claim.
Response: We have revised our recommendation for additional further studies excluding the words rapid and precise. Page 12, Line 453-454.
Comment 2: The reported data extend the authors’ previous CSA investigations of PBMSC to platelets as emphasized in the abstract (line 15 – 18) and elaborated in the introduction (lines 54-55, 66-79). Previous work ((Malinow, Schuh et al. 2018, McDowell, Aulak et al. 2020, Kaczara, Przyborowski et al. 2021, van der Heijden, van de Wijer et al. 2021, Kim, Lee et al. 2022, Yasseen, Elkhodiry et al. 2022, Esparza, Hernandez et al. 2023, Prakhya, Vekaria et al. 2023) excluding this reviewer) has already employed the CSA to evaluate platelet sources or preparation methods and should be adequately reflected in a revised version of the manuscript including the abstract, the introduction and the references.
Response: Previous studies have been adequately reflected in the revised version of the manuscript including in the abstract, introduction, and references. Page 1, Lines 15-18, page 2, Lines 55-58, 59-62, 85-92, and in the reference part.
Comment 3: The study population is heterogeneous: it comprises healthy individuals as well as patients and includes a wide age range (10-90 years). There is ample evidence that age, disease and medication impact the bioenergetics of platelets. The authors should address how these characteristics (mentioned in lines 333- 335, but not reported) could have affected the differences they observed between the conventional and the Optiprep platelet preparations without matching for such confounders.
Response: We agree that those unmatched clinical and Socio-demographic characteristics of the study participants could have affected the observed results of CSA between the two platelet isolation methods. We have addressed this issue in the limitation of the study in the revised manuscript. Page 10, Lines 332-337.
Comment 4: The preparation of the platelets was completed within 24h after blood collection. What were the storage conditions for the whole blood? Did the duration of this storage affect the CSA parameters?
Response: The whole blood collected with CPDA anticoagulant was stored at room temperature until the platelet isolation procedure. The storage condition is indicated in the revised manuscript on Page 10, Lines 364-365. We tested the effect of different times of storage on CSA parameters in our laboratory. The CSA parameters were determined immediately after blood draw, 24h-, 48h-, and 72h after blood draw, Storage up to 48h hours at room temperature does not have a major effect on CSA parameters results. A systematic review and Meta-analysis of 35 published articles on the effect of blood storage at room temperature on platelet-related measurements showed as well that there were no differences when tested 2 days later (https://doi.org/10.1016/j.ebiom.2017.09.024).
Comment 5: At least the Seahorse XF96 Analyzer (applied in the study for 25 out of 125 subjects) should be adequate to simultaneously analyse the bioenergetics of two platelet preparations from identical blood samples in multiples and thus reliably detect differences between both preparation methods. It would also be of interest to assess the platelet yield of both preparation methods. Do the platelet size distributions (as assessed by haematology counters) of the isolates differ between the methods? Could such differences (and the subpopulations) at least partially explain the differences of CSA parameters for different isolation methods?
Response: We compared different platelet isolation methods with regard to their effects on cytokine and growth factors release, on platelet size distributions (in our clinical chemistry department) and on platelet reaction to collagen and thrombin activation. We questioned whether the preparation method for plated-rich plasma, which is used therapeutically, influences platelet physiology and the therapeutical effect. The two platelet isolation methods described in the manuscript showed similar platelet size distribution. Identical numbers of platelets were used in the Seahorse analyzer independent of the type of platelet isolation method.
Comment 6: Moreover, platelets are cells/fragments which circulate in blood only for a few days and usually respond strongly to external signals but may also recover in vivo from such activation. How are these physiological dynamics reflected by the CSA results? Does the plate adhesion to the CSA well/plate activate platelets (instead of continuous flow in blood vessels, e.g. mimicked by stirring of a platelet suspension in a respirometer) and thereby affect respiration parameters?
Response: Platelets from both preparation methods were put in the same Seahorse analyzer plate. If platelet adhesion to the CSA wells activates platelets then platelets would be activated independent of their isolation method.
Comment 7: In their results (line 136) the authors conclude that platelets prepared by the Optiprep procedure were activated. They should include this interpretation in their Discussion section: the term “activated platelets” is usually considered in contrast to “normal”, i.e. resting, platelets. The authors should also discuss if platelet adhesion to the CSA wells independently of the preparation method could further contribute to platelet activation and how platelet activation in general may affect the relevance of CSA findings.
Response: The interpretation of platelet activation by Optiprep is included in the discussion. Page 9, lines 281 – 287.
We used the same CSA wells for CSA parameters measurement from platelets isolated by both methods. Therefore, if platelet adhesion to the CSA wells independently of the preparation method could further contribute to platelet activation, this effect will be the same for both isolation methods and this could not affect our observation and conclusion.
Comment 8: Are the authors really convinced that a correlation coefficient of only 0.44 despite its statistical significance allows for the meaningful conclusion that whole-cell ATP levels might be a sufficient marker to determine the “bioenergetic fitness in platelets (Figure 3) (lines: 220 -222)? Or does this statement refer to the correlation between basal respiration and platelet ATP levels for the two preparations which is not presented in a Figure? How does this conclusion corroborate the observation that the whole-cell (predominantly platelet) ATP did not differ between platelets prepared by the different methods (lines 104/5) despite significant differences between these preparations both in the basal respiration and in the spare capacity determined by CSA?
Response: We agree with the reviewer’s comment and removed this conclusion from the revised manuscript. Page 8, Lines 223 – 225.
Minor points:
Comment 1: Please use either “optiprep” or “Optiprep” consistently in text, tables and figures.
Response: Corrected using Optiprep consistently throughout the manuscript.
Comment 2: Figure 1: please replace “Tim” by “Time” in panel A and B and increase font size to discriminate the condition “with/without Optiprep”).
Response: Corrected as per the comment. Figure 1.
Comment 3: Line 123: replace “Tabel” by “Table”
Response: Corrected for Table 1 and Table 2.
Comment 4: Line 148: replace “50 platelets” by “50 platelet preparations” and “75 platelets” accordingly
Response: Corrected as Data were collected from 50 platelet preparations with Optiprep and 75 platelet preparations without Optiprep. Page 5, Lines 167-168
Comment 5: Figures 3 and 4: ggplot procedures usually scale plots according to the range of values. However, to facilitate direct comparison, please ensure identical scales to represent data for both preparations or combine the data from both platelet preparations into a single plot (using different symbols for the data and different colours/line markers for the regression line). Explain the representation of their confidence intervals (95%?) in the legend.
Response: In Figures 3 and 4, we intend to show the correlation between spare capacity and Intracellular ATP content (Figure 3), and between non-mitochondrial respiration and basal respiration, maximal respiration, or spare capacity (Figure 4) in platelets isolated by the two platelet isolation methods. In both figures, we have explained the correlation coefficients for platelets isolated using the two methods in the revised manuscript. Pages 6-8. Lines 207 -265.
Comment 6: 4.3 platelet isolation:
Lines 354 to 364 describe the conventional preparation method:
Response: We have clarified and corrected the description of the conventional platelet preparation method. Pages 10 and 11, Lines 368-377.
Comment 7: The sentence “The washing step was repeated without Optiprep” may be misleading in this context as it suggests that the first washing step included Optiprep? Please clarify.
Response: Corrected by removing the phrase “without Optiprep” from this sentence. Page 10, Line 375. Furthermore, we clarified and corrected the descriptions of both platelet isolation methods.
Comment 8: Lines 366 to 378 describe the Optiprep preparation:
Response: We have clarified and corrected the description of the Optiprep platelet preparation method. Page 11, Lines 384-407.
Comment 9: Line 372: supplementation of RPMI with glucose and pyruvate
Does this supplemented medium correspond to the “full medium” (line 374)? Was it also used for washing step 2? Why was it not used for the washing step(s) in the conventional preparation?
Response: Thank you very much for your comment. Supplementation of RPMI with glucose and pyruvate correspond to the “full medium”. We clarified this point. Moreover we clarified and corrected the platelet isolation methods. Both platelet isolation methods only differ with regard to the first step – the use of Optiprep versus centrifugation without Optiprep. The other steps were performed identically using the same medium and identical washing steps.
Comment 10: 4.6 Statistical Analyses (replace by “Statistical analyses”)
Two publications (Yepez, Kremer et al. 2018, Zhang, Yuan et al. 2021) have recommended specific statistical approaches to deal with outliers and variations of the CSA. Please describe if and how the Wave 2.6.0. software and your cleaning of the data in Excel addressed these methodological issues.
Response: Statistical analyses replaced by Statistical analyses. Page 12, line 433.
We used Excel to clean the data meaning to avoid errors during data entry (wrong data entry, missing data, duplication of data during entry) before the data analyses using R program but not to clean the outliers and variations of the CSA data. We have edited the text to avoid such confusion. Page 12, Line 436.
Round 2
Reviewer 1 Report
Comments and Suggestions for Authors
The authors have addressed the questions and concerns raised in the first review
Author Response
Dear reviewer, thank you very much once again for your review of our revised manuscript and for your useful comments in the first round. Your comments and suggestions helped us to improve the quality of our manuscript significantly.
Reviewer 2 Report
Comments and Suggestions for Authors
The revised version of the ms addresses most of the comments by the reviewer. A few issues remain before I can recommend to accept a final version.
Comments adressing major issues
comment 4: In their response the authors share that they assessed the CSA changes observed during storage of the whole blood (identical samples?) for 0, 24, 48 and 72 h after collection. They should provide these results in the ms as they reflect a critical aspect of the method.
The metaanalysis cited in their response appears not to be relevant as the selected parameters are conventional (cell counts and clinical chemistry) but not valid indicators of metabolic activity.
comment 5: The authors refer to their analysis of numerous parameters. With the exception of the size distribution, it remains open if significant differences were observed between the two preparation methods (using identical blood samples?)?
comment 9: please add the concentration of PGI2
references 8, 26: authors' names are all in capital letters
Author Response
Reviewer 2 Report 2
Review Report Form
Open Review
(x) I would not like to sign my review report
( ) I would like to sign my review report
Quality of English Language
( ) I am not qualified to assess the quality of English in this paper
( ) English very difficult to understand/incomprehensible
( ) Extensive editing of English language required
( ) Moderate editing of English language required
( ) Minor editing of English language required
(x) English language fine. No issues detected
|
Yes |
Can be improved |
Must be improved |
Not applicable |
|
|
Does the introduction provide sufficient background and include all relevant references? |
(x) |
( ) |
( ) |
( ) |
|
Are all the cited references relevant to the research? |
(x) |
( ) |
( ) |
( ) |
|
Is the research design appropriate? |
( ) |
(x) |
( ) |
( ) |
|
Are the methods adequately described? |
(x) |
( ) |
( ) |
( ) |
|
Are the results clearly presented? |
( ) |
(x) |
( ) |
( ) |
|
Are the conclusions supported by the results? |
(x) |
( ) |
( ) |
( ) |
Comments and Suggestions for Authors
General comment: The revised version of the ms addresses most of the comments by the reviewer. A few issues remain before I can recommend to accept a final version.
Response: Dear reviewer, thank you so much once again for your review of our revised manuscript and for providing us with very useful comments. The manuscript has been revised according to your comments to improve its quality.
Comments adressing major issues
Comment 4: In their response the authors share that they assessed the CSA changes observed during storage of the whole blood (identical samples?) for 0, 24, 48 and 72 h after collection. They should provide these results in the ms as they reflect a critical aspect of the method.
The metaanalysis cited in their response appears not to be relevant as the selected parameters are conventional (cell counts and clinical chemistry) but not valid indicators of metabolic activity.
Response: We assessed the CSA changes during storage of the whole blood for 0, 24, 48 and 72 h after collection. We used identical samples. The same procedure we did as well for analysis of CSA parameters in peripheral mononuclear blood cells (PBMC). Up to 48h after blood draw no significant changes in CSA parameters were observed. We included this information into the MS. Page 10, lines 366-368.
Comment 5: The authors refer to their analysis of numerous parameters. With the exception of the size distribution, it remains open if significant differences were observed between the two preparation methods (using identical blood samples?)?
Response: We started our analysis on CSA parameters within PBMC using Optiprep. Optiprep was choosen due to the guaranted low endotoxin concentration. Optiprep was chosen at a density of 1.077g/ml for PBMC isolation. Another reason to select Optiprep for cell separation was to remove platelets from the PBMC fraction. Due to a different metabolism of platelets we were aware that platelet contamination (dependent on the number of contaminating platelets) might significant influence the CSA parameters of PBMC. Thus, Optiprep isolateion at 1.077g/ml was followed by an Optiprep isolation at a density of 1.063g/ml. Using this procedure we could reduce the amount of contaminating platelets down to 5-10%. Another reason for choosing Optiprep was the ability to isolate several different cell types one after the other from the same blood sample, such as first PBMC and then platelets using the two consecutive centrifugation steps (Optiprep at densities of 1.077 and 1.063g/ml). In order to further reduce the contamination with platelets in the PBMC fraction so that any influence on the CSA parameters could be ruled out, we changed our processing. We decided to process the PBMC using negative selection (StemCell). From this point on, we changed our platelet isolation method. We worked on the new platelet isolation method comparing the lieterature. Finally, we compared both methods with an identical blood sample. We performed the comparisons only with a small number of samples. The results were inconsistent. We obtained identical results in the comparisons, results with less deviation and results with higher deviation. These initial inconsistent results when comparing both methods directly with a blood sample prompted us to write this manuscript.
comment 9: please add the concentration of PGI2
Response: The final concentration of PGI2 was 0.5 µM. We included that information into the MS. Page 11, lines 379 and 394-395.
References 8, 26: authors' names are all in capital letters
Response: Corrected